# Unfavorable Dietary Quality Contributes to Elevated Risk of Ischemic Stroke among Residents in Southwest China: Based on the Chinese Diet Balance Index 2016 (DBI-16)

**DOI:** 10.3390/nu14030694

**Published:** 2022-02-07

**Authors:** Yingying Wang, Xu Su, Yun Chen, Yiying Wang, Jie Zhou, Tao Liu, Na Wang, Chaowei Fu

**Affiliations:** 1Key Laboratory of Public Health Safety, NHC Key Laboratory of Health Technology Assessment, School of Public Health, Fudan University, Shanghai 200032, China; 17211020095@fudan.edu.cn (Y.W.); 18211020001@fudan.edu.cn (Y.C.); fcw@fudan.edu.cn (C.F.); 2Guizhou Province Center for Disease Prevention and Control, Guiyang 550004, China; susuxuxu@163.com (X.S.); wyy123789123789@163.com (Y.W.); zhoujie19872014@163.com (J.Z.)

**Keywords:** dietary quality, ischemic stroke, Chinese Diet Balance Index 2016, DBI-16, Southeast China

## Abstract

Background: Little is known about the effects of dietary quality on the risk of ischemic stroke among Southwest Chinese, and evidence from prospective studies is needed. We aimed to evaluate the associations of ischemic stroke with dietary quality assessed by the Chinese Diet Balance Index 2016 (DBI-2016). Methods: The Guizhou Population Health Cohort Study (GPHCS) recruited 9280 residents aged 18 to 95 years from 12 areas in Guizhou Province, Southwest China. Baseline investigations, including information collections of diet and demographic characteristics, and anthropometric measurements were performed from 2010 to 2012. Dietary quality was assessed by using DBI-2016. The primary outcome was incident ischemic stroke diagnosed according to the International Classification of Diseases 10th revision (ICD-10) until December 2020. Data analyzed in the current study was from 7841 participants with complete information of diet assessments and ischemic stroke certification. Cox proportional hazards models were used to estimate the risk of ischemic stroke associated with dietary quality. Results: During a median follow-up of 6.63 years (range 1.11 to 9.53 years), 142 participants were diagnosed with ischemic stroke. Participants with ischemic stroke had a more excessive intake of cooking oils, alcoholic beverages, and salt, and had more inadequacy in meats than those without ischemic stroke. (*p* < 0.05). Compared with participants in the lowest quartile (Q1), those in the highest quartile (Q4) of the higher bound score (HBS) and of the dietary quality distance (DQD) had an elevated risk for ischemic stroke, with the corresponding hazard ratios (HRs) of 3.31 (95%CI: 1.57–6.97) and 2.26 (95%CI: 1.28–4.00), respectively, after adjustment for age, ethnic group, education level, marriage status, smoking and waist circumference, and the medical history of diabetes and hypertension at baseline. In addition, excessive intake levels (score 1–6) of cooking oils, excessive intake levels (score 1–6) of salt, and inadequate intake levels (score −12 to −7) of dietary variety were positively associated with an increased risk for ischemic stroke, with the multiple HRs of 3.00 (95%CI: 1.77–5.07), 2.03 (95%CI: 1.33–3.10) and 5.40 (95%CI: 1.70–17.20), respectively. Conclusions: Our results suggest that unfavorable dietary quality, including overall excessive consumption, excessive intake of cooking oils and salt, or under adequate dietary diversity, may increase the risk for ischemic stroke.

## 1. Introduction

Stroke is the second leading cause of death and disability worldwide, and has been the first leading cause of death and the leading cause of all-age disability-adjusted life years (DALYs) in China [1,2]. Notably, nationwide studies and periodic governmental reports revealed a great burden of stroke in China, with an increasing prevalence and incidence in the past decade [3,4]. According to the latest annual report in 2019, China currently has 21 million patients with stroke [5]. Ischemic stroke, defined as the permanent infarction of cerebral tissues due to abrupt decreases in cerebral blood flow, accounts for more than 70% of prevalences among all sub-types of stroke. A large prospective cohort study based on half a million Chinese adults showed that the incidence of ischemic stroke during 7.2-year follow-up was 5.86 per 1000 person-years [6].

Dietary intake is a modifiable lifestyle behavior closely associated with most non-communicable diseases (NCDs), including cardiovascular diseases (CVDs) [7,8,9]. There has been an increasing interest in using specific indexes to evaluate dietary quality and their effects on NCDs, especially in some developed countries, such as the Healthy Eating Index (HEI) and the Diet Quality Index (DQI) developed for Americans [10,11], and the Mediterranean Diet Score (MDS) used for the residents in Northern Europe [12]. With reference to the methods of HEI and DQI, the Chinese Dietary Balance Index (DBI) has been designed to assess the overall diet quality in the Chinese population [13], according to the most recent Dietary Guidelines for Chinese residents [14]. Although associations of DBI with diabetes, hypertension, and cardiometabolic risk factors have been reported in previous cross-sectional studies among subgroups in China [15,16,17], data on the relationship between DBI and ischemic stroke is inconclusive, and evidence from prospective studies are warranted.

In this current study, we aimed to explore the associations between diet quality assessed by DBI and risk of incident ischemic stroke based on a prospective cohort in Guizhou Province in Southwest China, a region where economy and culture are relatively underdeveloped and with a great disease burden of ischemic stroke [18], in order to provide some evidence on further dietary intervention to manage and prevent ischemic stroke.

## 2. Methods

### 2.1. Study Design and Participants

The Guizhou Population Health Cohort Study (GPHCS) is one of few large population-based prospective cohort studies in Southwest China, which was established from 12 areas (five urban districts and seven rural counties) in Guizhou Province between 20 November 2010 and 19 December 2012. A multistage proportional stratified cluster sampling method was used to obtain a representative sample of the general population in Guizhou Province. The inclusive criteria were as follows: (1) age of 18 years or above; (2) living in the study regions for more than six months and having no plan to move out; (3) completing survey questionnaire and blood sampling; (4) signing the written informed consent. A total of 9280 local residents were enrolled in the cohort. All participants were followed up for major chronic diseases and vital status through a repeated investigation conducted between 2016 and 2020. All deaths were confirmed by the record from Death Registration Information System and Basic Public Health Service System. Ethics approval was obtained from the ethics review board of Guizhou Province (No.S2017-02). All participants provided written informed consent at enrollment.

In this study, we excluded participants with a history of ischemic stroke, haemorrhagic stroke, myocardial infarction or other cardiovascular diseases, missing data of diet consumption or cardiovascular diseases at baseline, loss to follow-up, and death, leaving 7841 participants for the analyses (Figure 1).

### 2.2. Outcome Definition

The primary outcome was the first onset of ischemic stroke (I63) diagnosed according to the International Classification of Diseases 10th revision (ICD-10). All reported events were reviewed and integrated centrally by trained clinical staff. Each participant was followed up until the first occurrence of the corresponding outcome, death, or loss to follow-up, which occurred first before 31 December 2020. The incidence rate was calculated as the number of incident cases divided by follow-up person-years.

### 2.3. Dietary Data Collection

The information of dietary intake for each participant was assessed using a semi-quantitative food frequency questionnaire (FFQ), both at the individual level and at the household level. The individual FFQ covered 23 items of foods and beverages which were commonly consumed, including cereals, tubers, livestock meats, poultry meats, aquatic products, vegetables, fruits, eggs, dairy products, soybean products, etc. A commonly used unit or portion size was specified for each food item, participants were required to answer their usual consuming frequency (daily, weekly, monthly, yearly, or never) of each specific food or beverages over the past one year and the amount of consumption at each time. The daily intake on average for each food item was then calculated according to the product of the intake frequency and the amount consumed at each time (in gram per day, g/day). House condiment consumption, such as cooking oils, salt, sugar, sauces, etc., was determined by evaluating all condiments consumed by all household members for one month. The total amount of condiments consumed in the household divided by the number of members usually eating at home was used to assess individual consumption of condiments.

### 2.4. Dietary Intake Assessment

The dietary quality among the participants at baseline was assessed by the Chinese Diet Balance Index 2016 (DBI-16) [13], a revised version from the Chinese Diet Balance Index 2007 (DBI-07) [19]. DBI-2016 comprises 14 subgroups of 8 components from the Dietary Guidelines for Chinese residents [14], including: (1) cereal; (2) vegetable and fruit; (3) dairy and soybean; (4) animal food (red meats/products/poultry/game, fish/shrimp, and egg); (5) empty energy foods (cooking oils, and alcoholic beverage); (6) condiments (addible sugar, and salt); (7) diet variety; and (8) drinking water. A score of 0 for each DBI-16 component means that the individual has reached the recommended intake amounts of the corresponding food group. Positive scores (ranging 1 to 12) indicate the excessive intake level of cereals, red meat/products/poultry/game, eggs, cooking oils, alcoholic beverage, addible sugar, salt, while negative scores (ranging −12 to −1) indicate the inadequate intake level of cereals, vegetables, fruits, dairy, soybeans, red meat/products/poultry/game, fish/shrimps, eggs, diet variety, and drinking water. Considering the difference of nutrient requirements in energy consumption, the scoring of these 14 food subgroups was based on 11 levels of energy intake. Scoring details of DBI-16 are shown in Appendix A.

Based on the scores for each DBI-16 component, three indicators of diet quality were calculated: (1) the lower bound score (LBS), an indicator for inadequate food intake, was computed by adding all the negative scores; (2) the higher bound score (HBS), an indicator for excessive food intake, was calculated by adding all the positive scores; (3) the diet quality distance, an indicator of unbalanced food intake, is calculated by adding the absolute values of both positive and negative scores [13]. The ranges of LBS, HBS, and DQD were 0 to 60, 0 to 40, and 0 to 84, respectively. For simplicity, each indicator was further divided into five levels to reflect the diet quality: (1) no problem, a score of 0; (2) almost no problem, less than 20% of the total score; (3) low-level problem, between 20% and 40% of the total score; (4) moderate level problem, between 40% and 60% of the total score; and (5) high-level problem, greater than 60% of the total score.

### 2.5. Other Variables Collection

A standardized questionnaire was used to collect the information of demographic characteristics, lifestyles, and medical history, including age, sex, area, ethnic group, education level, family income, marriage status, occupation status, physical activity, smoking or not, alcohol drinking or not, medical history of diabetes, hypertension and dyslipidemia, use of medications and nutraceuticals. Smoking was defined as smoking at least one cigarette a day for 12 months or more. Alcohol drinking was defined as drinking at least three times a week for 12 months or more. Medication use was defined as taking medications for diabetes, hypertension, dyslipidemia, or obesity regularly. Nutraceutical intake was defined as intaking some common nutraceuticals (such as vitamins or minerals), or foods with health-care functions (such as wine, tea) at least one time a week for 12 months or more. The physical activity level was calculated as the product of the duration and frequency of each activity, weighted by an estimate of the metabolic equivalent (MET) of that activity and summed for all activities performed, with the result expressed as the average MET hours per day.

Height, weight, and waist circumference were determined by trained technicians, using calibrated instruments with standard protocols and recorded to the nearest 0.1 cm or 0.1 kg. Waist-to-height ratio (WHtR) was calculated as waist circumference in centimeters divided by height in meters. Body mass index (BMI) was calculated as weight in kilograms divided by the square of height in meters. Systolic blood pressure (SBP) and diastolic blood pressure (DBP) were measured from the left arm after the participant rested in a seated position. All participants provided a 10-mL blood sample after an overnight fast of at least 10 h, they were also required to undergo an oral glucose tolerance test (OGTT), and the plasma was obtained at 2 h during the test. Fasting plasma glucose (FPG), 2-h postload glucose (2h-PG) and Hemoglobin A1c (HbA1c) were determined by the glucose oxidase methods (Roche Diagnostics, Mannheim, Germany). Serum triglycerides (TG), total cholesterol (CHOL), low-density lipoprotein cholesterol (LDL-C), and high-density lipoprotein cholesterol (HDL-C) were measured by enzymatic methods (Roche Diagnostics, Mannheim, Germany).

Diabetes was defined as those above the threshold of glycemia (FPG ≥ 6.1 mmol/L or 2h-PG ≥ 7.8 mmol/L), having a reported diabetes history, or experiencing anti-diabetes medications [20]. Hypertension was defined as an abnormal level of current blood pressure (SBP > 140 mmHg or DBP > 90 mmHg), having a reported hypertension history, or experiencing anti-hypertension medications [20]. Dyslipidemia was defined as an abnormal level of current blood lipids (TG ≥ 1.7 mmol/L, CHOL ≥ 5.2 mmol/L, LDL ≥ 3.4 mmol/L, HDL < 1.0 mmol/L), having a reported dyslipidemia history, or experiencing anti-dyslipidemia medications [20]. General overweight or obesity was defined as BMI ≥ 24 kg/m^2^, central obesity was defined as WC ≥ 85 cm for females or ≥ 90 cm for males, and obesity status was defined as having either of these two types of obesity [21].

### 2.6. Statistical Analysis

Continuous variables were expressed as means and standard deviations (mean ± SD) and compared by using the Student’s *t*-test. Categorical variables were presented as frequencies and percentages (*n*, %) and compared by using the Chi-square test. Considering that the proportional hazards assumption showed no strong evidence of departure, cox proportional hazards models were used to estimate hazard ratios (HRs) and 95% confidence intervals (95%CIs) for ischemic stroke by the components of DBI-16 and the indicators of diet quality. The level of statistical significance was defined as α = 0.05 of two-side probability. All analyses were performed using the R program (version 4.0.4, R Foundation for Statistical Computing, Vienna, Austria), and all figures were performed by using GraphPad Prism software (version 9, GraphPad Prism, San Diego, CA, USA).

## 3. Results

### 3.1. Descriptions of Study Population

The baseline characteristics of all 7841 participants in this study are shown in Table 1. The mean age was 44.18 ± 14.97 years at enrollment, and 47.4% (*n* = 3719) were male. Of these, 67.1% (*n* = 5258) of participants were rural residents, and more than half were Han Chinese (58.5%, *n* = 4589) and farmers (57.3%, *n* = 4490). The majority had an education level below junior middle school or no formal educated (86.7%, *n* = 6799).

During a median follow-up of 6.63 years (range 1.11 to 9.53 years), 142 participants were diagnosed with ischemic stroke. Compare with participants without ischemic stroke, those with incident ischemic stroke were more likely to be older, Han Chinese, with lower economic level, and less likely to be formally educated, and married (divorced/widowed/separated). Those with ischemic stroke also tended to have prevalent diabetes and hypertension (*p* < 0.001).

### 3.2. Assessments of Dietary Quality

The distributions of scores for the DBI-16 components are presented in Table 2. Overall, 0.3% to 98.9% of participants have reached the recommended dietary intakes (score = 0) of the DBI-16 components, and the majority (over 90%) consumed appropriate amounts of addible sugar and alcoholic beverages. Inadequate intakes (score < 0) were most commonly observed in dairy, fish, fruits, eggs, vegetables and soybeans, with the corresponding proportions among all participants of 99.7%, 97.5%, 95.5%, 83.5%, 62.0%, and 54.4%, respectively. Over 84.8 % of individuals had a dietary variety below the recommended level. By contrast, excessive intakes (score > 0) in cereals, cooking oils, salt and meats were also observed among 72.5%, 64.1%, 60.8%, and 46.6% of participants, respectively. Participants with ischemic stroke had more excessive intake in cooking oils, alcoholic beverages, and salt, and were more likely to have inadequate intake in meats than those without ischemic stroke. (*p* < 0.05).

The DBI-16 also revealed that 57.3%, 35.1%, and 2.3% of participants had a low, moderate, and high level of under intake (indicated by LBS), respectively; 43.8%, 16.8%, and 0.9% of them had a low to high level of over intake (indicated by HBS), respectively; 50.6%, 44.1%, and 3.9% of them had a low to high-level problem of overall unbalance (indicated by DQD), respectively (Table 3). The ischemic stroke patients had a higher median HBS, higher prevalence of moderate level of over intake (HBS, 25.4%) and higher overall unbalance (DQD, 48.6%) as compared with those without ischemic stroke. Moreover, the distributions of dietary quality across two groups divided by area and ethnicity are shown in Figure 2.

### 3.3. Association Analyses of Ischemic Stroke with Dietary Quality Indicators and DBI-16 Components

The results of Cox regression analyses were shown in Table 4. The hazard ratios (HRs) for ischemic stroke were progressively elevated with increasing HBS. Compared with participants in the lowest quartile (Q1) of HBS, those in the highest quartile (Q4) had a 3.15-fold (95%CI: 1.50–6.63) increased risk for ischemic stroke, after adjustment for age, sex, area, ethnic group, education level, marriage status, smoking, diabetes, hypertension, dyslipidemia, and obesity status; (Model 2), and the association slightly increased after additional adjustment for medication use and nutraceutical intake at baseline (HR: 3.31, 95%CI: 1.57–6.97, Model 3). A similar result was observed for the highest DQD quartile (Q4), with the corresponding multiple-adjusted HR of 2.19 (95%CI: 1.24–3.86) based on Model 2 and 2.26 (95%CI: 1.28–4.00) based on Model 3.

Among the thirteen components of DBI-16, both cooking oils and salt showed significant associations with ischemic stroke. Compared with the appropriate intake level (score = 0), excessive intake level (score 1–6) in cooking oils or salt added a 200% risk (HR: 3.00, 95%CI: 1.77–5.07, Model 3) and 103% risk (HR: 2.03, 95%CI: 1.33–3.10, Model 3) for ischemic stroke, respectively. Moreover, lower dietary variety (score −12 to −7) also promoted incident ischemic stroke (HR:5.40, 95%CI: 1.70–17.20, Model 3).

### 3.4. Stratified Analyses of the Association of Ischemic Stroke with Dietary Quality Indicators across Different Status of Comorbidities, Medication Use, and Nutraceutical Intake

After eliminating the role of comorbidities, medications, and nutraceuticals, HBS was still related to ischemic stroke among participants without diabetes or obesity, or those free of nutraceuticals or medications (Figure 3). In addition, the effect strengths of the associations between HBS with ischemic stroke were more evident in those with hypertension history (HR: 7.10, 95%CI: 2.71–19.9) and using medications (HR: 6.30, 95%CI: 1.44–18.6) than in total participants, comparing the highest quartile (Q4) of HBS to the lowest quartile (Q1) based on Model 3. However, HBS seemed to play less of a role in ischemic stroke among those with diabetes or intaking nutraceuticals. A similar pattern was also observed for DQD, with the increased HRs of 4.87 (95%CI: 1.66–14.20, Q4 vs. Q1, Model 3) and 6.00 (95%CI: 1.39–17.70, Q4 vs. Q1, Model 3) among those with hypertension history and using medications, respectively.

### 3.5. Stratified Analyses of the Associations of Ischemic Stroke with Dietary Quality Indicators across Baseline Demographic Factors

The multi-adjusted HRs for ischemic stroke by dietary quality indicators varied according to different demographic factors (Figure 4). The positive associations between HBS and ischemic stroke were seen only among participants with a baseline age of 60 years or more (HR: 4.70, 95%CI: 2.89–16.00, Q4 vs. Q1), female (HR: 2.94, 95%CI: 1.17–7.34, Q4 vs. Q1), rural residents (HR: 3.50, 95%CI: 1.57–7.81, Q4 vs. Q1) and the Ethnic Han (HR: 3.15, 95%CI: 1.36–7.32, Q4 vs. Q1), although there was no significant effect modification (*p*_interaction_ > 0.05) by age, sex, area, and ethnic group. When predicted by DQD, the risks for ischemic stroke elevated in females (HR: 2.68, 95%CI: 1.24–5.80, Q4 vs. Q1), rural residents (HR: 2.49, 95%CI: 1.26–4.90, Q4 vs. Q1), and the Ethnic Han (HR: 2.80, 95%CI: 1.36–5.77, Q4 vs. Q1). There was additionally a negative association between LBS and ischemic stroke in those aged more than 60 years (HR: 0.12, 95%CI: 0.03–0.55, Q3 vs. Q1).

## 4. Discussions

In this prospective cohort study conducted in Guizhou Province, Southwest China, we observed that the participants were exposed to dietary unbalance to different extents at baseline, mainly including the inadequate intakes of dairy, fish, fruits, eggs, vegetables, and soybeans, and the excessive intakes of cereals, cooking oils, salt, and meats. Our analyses further suggested that unfavorable dietary quality, including overall excessive consumption, high oils and salt diet, and low food diversity, may be a risk for ischemic stroke.

Several studies conducted in American, Swedish, and Chinese populations have estimated the effects of dietary quality on ischemic stroke. A recent study based on 73,890 women in Nurses’ Health Study (NHS, 1984–2016), 92,352 women in NHSII (1991–2017), and 43,266 men in Health Professionals Follow-Up Study (1986–2012) in America revealed that the healthful plant-based dietary quality assessed by the Plant-based Diet Index (PDI) seemed to reduce the risk of ischemic stroke [22]. Similarly, a prospective cohort study of 26,547 Swedish aged 46 to 73 years found that less risk of incident ischemic stroke was related to the higher dietary quality, which was based on adherence to the Swedish nutrition recommendations [23]. By contrast, a large-scale prospective cohort study of 512,725 Chinese aged 30 to 79 years reported that less healthy dietary habits, which was defined as non-daily eating of vegetables, fruits, and eggs combined with eating daily or less than weekly, contributed to incident ischemic stroke [24].

China has experienced an ongoing transition of dietary patterns over the past decades, which mainly featured declines in the intakes of coarse and refined grains and vegetables, as well as increases in the intakes of animal-derived foods, with pork being most popular. Intakes of eggs, fish, and dairy have been consistently below recommended levels [25]. In addition, as the cooking methods markedly shifted from steaming and boiling to stir-fried and deep-fried, the daily consumption of cooking oils has been gradually increased from 18.2 g to 42.1 g from 1982 to 2012 [26]. People preferred to add a certain amount of salt to keep food from spoiling, such as pickles and salted fish, especially in rural or remote areas without a good condition for food storage. Data from China National Nutrition Surveys (CHNS) also revealed that more than 55.9% and 71.8% of Chinese have consumed excessive cooking oil and salt, which were far above the recommended levels and strongly linked to increased risk of chronic diseases [27]. These cooking and eating behaviors are partly driven both by the great accessibility of cooking oils and salt and by their low price.

Compared to a previous study from Shanghai in East China and a cross-sectional study based on the CHNS, participants in this current study from Guizhou Province in Southeast China have a higher percentage of excessive intake of cooking oil and salt (score > 0) [28,29]. We also observed that the probabilities of occurring ischemic stroke for participants consuming the excessive level of cooking oil and salt were about appropriate two to three times as high as those consuming the moderate level. Cooking oil has been the main source of fat intake these decades. As expected, a diet with elevated fat or salt was associated with a significantly increased risk of hypertension and stroke, as reported previously [30,31]. A rat experiment found that a long-term administration of canola oil, sesame oil, or trans-fat led to marked dyslipidemia, fat accumulation, neuroinflammation, vascular lesion, and endothelial injury for rats, and thereby remarkably contributed to ischemic and hemorrhagic strokes [32]. Results from a gerbil animal model study indicated that a high-fat diet accelerates and exacerbates microgliosis and neuronal damage [33]. Although extra virgin olive oil (EVOO) or other oils rich in monounsaturated fat have been reported to bring some benefits to the cardiocerebral vascular system, but their consumption by the Chinese was relatively low [26]. In numerous epidemiology and clinical studies, excessive dietary sodium salt intake was linked to hypertension, which has been considered as the main risk factor for stroke [34]. A large prospective cohort study observed a significant linear association between calibrated urinary sodium excretion and stroke in patients with chronic kidney disease [35].

Given the potential effects of comorbidities, some medications for metabolic diseases, and some nutraceuticals, which may promote or decelerate the progress of cardiovascular diseases [36,37], we performed the same analyses in those with different status’ of these conditions. The risk effects of excessive food intake (evaluated by HBS) and unbalanced food intake were more evident in those with hypertension history and taking medications for metabolic diseases, indicating that people with a high risk of stroke should pay more attention to the balance of food types and daily intake, reasonable diet is also one of the major measures to prevent hypertension [16]. However, we failed to observe any associations of ischemic stroke with diet quality among those with diabetes or intaking nutraceuticals, perhaps due to a smaller sample size in those subgroups.

The current study provides primary evidence that the risk of ischemic stroke among the residents from Guizhou Province in Southwest China may be partly due to the potential effects of dietary quality. The chief strengths lie in the use of a large population-based cohort, a prospective study design with more than six years of follow-up, a dietary assessment index suitable for Chinese people. The scores of DBI-16 are based on different levels of energy consumption for specific individuals recommended by the most recent Dietary Guidelines for Chinese residents, so the potential confounding effects of total energy intake may be appropriately controlled [38]. There also exist certain limitations. Firstly, dietary habits and socioeconomic characteristics were collected based on individual self-report, which might lead to recall bias. The food frequency questionnaire (FFQ) has been considered a convenient and widely-used dietary assessment tool, but this method is subject to less accuracy of quantification of food portions than the method of weighing, which might make some measurement errors [39]. Secondly, we just applied the proportion of condiment intakes at home to estimate the total daily intakes, inevitably ignoring the condiment intakes from eating outside. Thirdly, our study population was from Southeast China and the dietary assessment index was applicable to the Chinese, so the findings from the current study should be generalized with caution to other populations. In conclusion, considering a great disease burden caused by ischemic stroke in China, our study suggests that it is essential to adjust dietary habits and conduct dietary interventions, especially controlling risk factors in preventive mode better than therapeutic mode, because of a fairly short time to death or irreversible injuries after the onset of stroke.

## 5. Conclusions

Our results suggest that unfavorable dietary quality, including overall excessive consumption, excessive intake of cooking oils and salt, or under adequate dietary diversity, may increase the risk for ischemic stroke.

## Figures and Tables

**Figure 1 nutrients-14-00694-f001:**
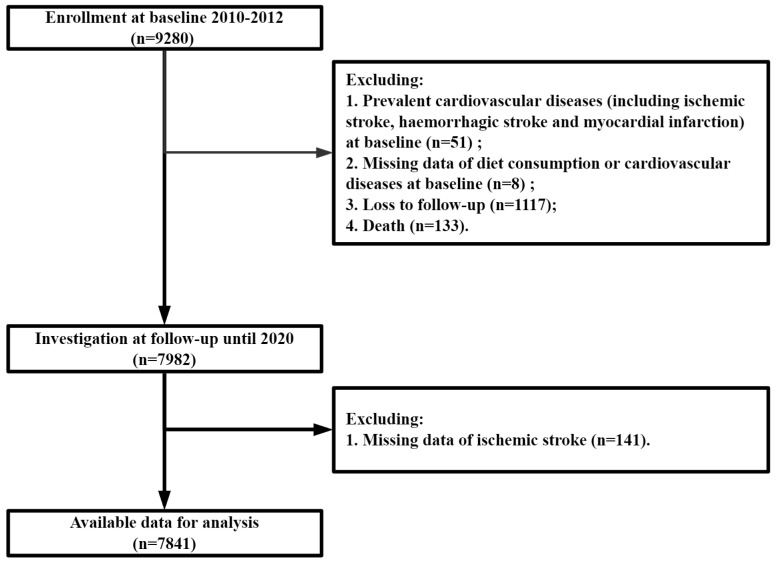
Flow chart of the study.

**Figure 2 nutrients-14-00694-f002:**
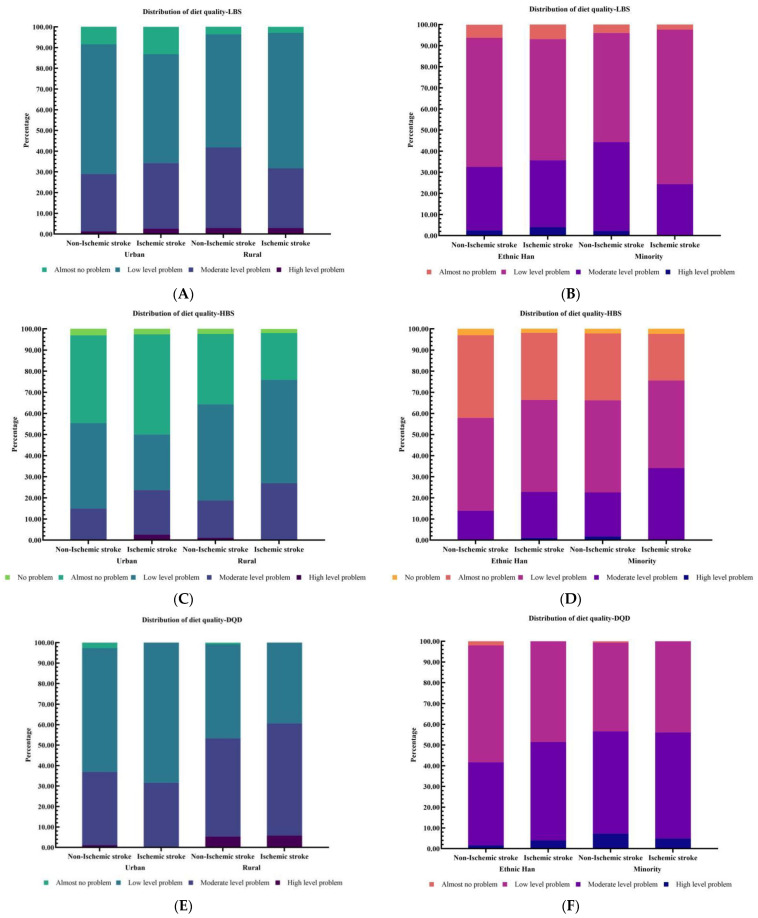
Distribution of the diet quality stratified by area (urban/rural) and ethnic group (ethnic Han/minority): (**A**,**B**) for the lower bound score (LBS); (**C**,**D**) for the higher bound score (HBS); (**E**,**F**) for the diet quality distance (DQD).

**Figure 3 nutrients-14-00694-f003:**
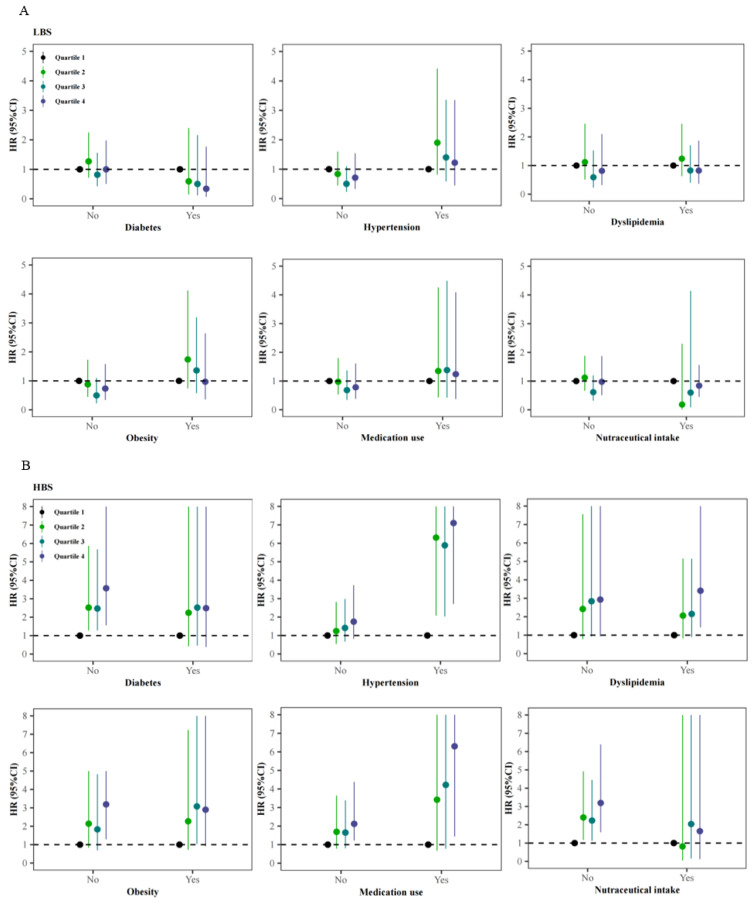
Adjusted hazard ratios (HRs) and 95% confidence intervals (95%CIs) for ischemic stroke associated with baseline dietary quality after stratified by the status of diabetes, hypertension, dyslipidemia, obesity, medication use and nutraceutical intake: (**A**) for the lower bound score (LBS); (**B**) for the higher bound score (HBS); (**C**) for the diet quality distance (DQD).

**Figure 4 nutrients-14-00694-f004:**
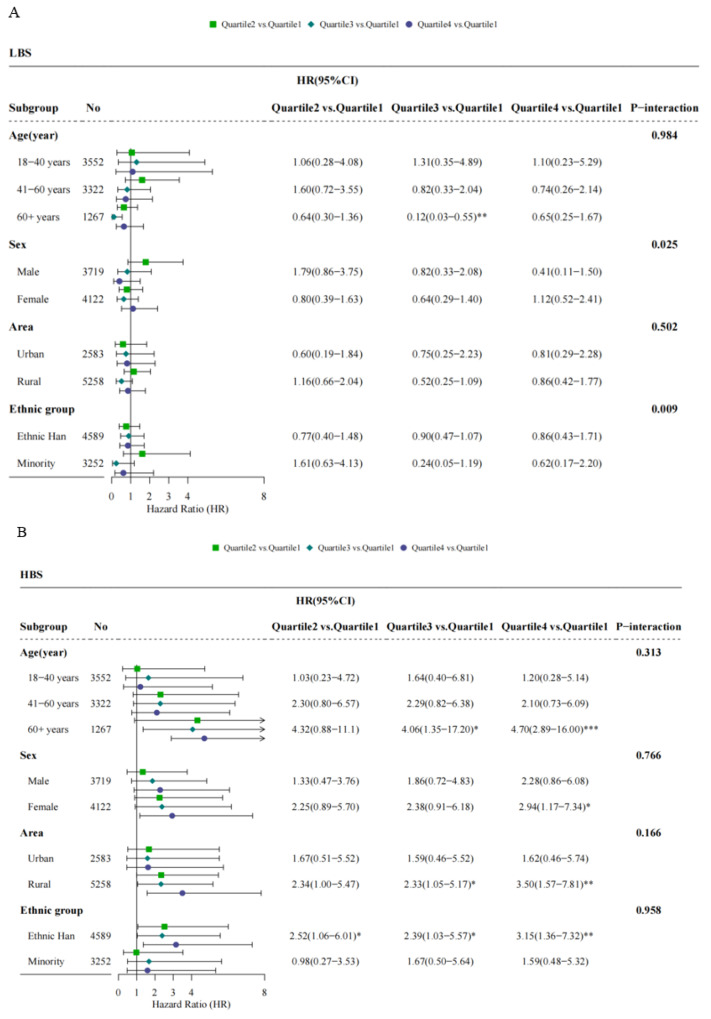
Adjusted hazard ratios (HRs) and 95% confidence intervals (95%CIs) for ischemic stroke associated with baseline dietary quality after stratified by age, sex, area, and ethnic group: (**A**) for the lower bound score (LBS); (**B**) for the higher bound score (HBS); (**C**) for the diet quality distance (DQD); * *p* < 0.05; ** *p* < 0.01; *** *p* < 0.001.

**Table 1 nutrients-14-00694-t001:** Baseline Characteristics of participants according to ischemic stroke status.

Characteristics	All (*n* = 7841)	Non-Ischemic Stroke (*n* = 7699)	Ischemic Stroke (*n* = 142)	*p* Value
Age (*n*, %)				<0.001
18–40 years	3252 (41.5)	3231 (42.0)	21 (14.8)	
41–60 years	3322 (42.4)	3256 (42.3)	66 (46.5)	
≥60 years	1267 (16.1)	1212 (15.7)	55 (38.7)	
Sex (*n*, %)				0.754
Male	3719 (47.4)	3654 (47.5)	65 (45.8)	
Female	4122 (52.6)	4045 (52.5)	77 (54.2)	
Area (*n*,%)				0.136
Urban	2583 (32.9)	2545 (33.1)	38 (26.8)	
Rural	5258 (67.1)	5154 (66.9)	104 (73.2)	
Ethnic group (*n*,%)				0.003
Ethnic Han	4589 (58.5)	4488 (58.3)	101 (71.1)	
Minority	3252 (41.5)	3211 (41.7)	41 (28.9)	
Education (*n*,%)				0.038
No formal education	1606 (20.5)	1565 (20.3)	41 (28.9)	
Junior middle school and below	5193 (66.2)	5111 (66.4)	82 (57.7)	
Senior high school and above	1042 (13.3)	1023 (13.3)	19 (13.4)	
Family income (*n*, %)				0.010
<3000 RMB/person	1664 (32.6)	1625 (32.6)	39 (33.9)	
3000–10,000 RMB/person	2129 (41.8)	2080 (41.7)	49 (42.6)	
≥10,000 RMB/person	1306 (25.6)	1279 (25.7)	27 (23.5)	
Marriage (*n*,%)				0.001
Married/Cohabit	6340 (80.9)	6226 (80.9)	114 (80.3)	
Unmarried/Single	744 (9.5)	740 (9.6)	4 (2.8)	
Divorced/Widowed/Separated	757 (9.7)	733 (9.5)	24 (16.9)	
Occupation (*n*, %)				0.284
Farmers	4490 (57.3)	4407 (57.2)	83 (58.5)	
Others	2092 (26.7)	2061 (26.8)	31 (21.8)	
Unemployed or retired	1259 (16.1)	1231 (16.0)	28 (19.7)	
Smoking (*n*, %)				0.375
No	5856 (74.7)	5755 (74.7)	101 (71.1)	
Yes	1985 (25.3)	1944 (25.3)	41 (28.9)	
Alcohol drinking (*n*, %)				0.403
No	6038 (77.0)	5924 (76.9)	114 (80.3)	
Yes	1803 (23.0)	1775 (23.1)	28 (19.7)	
Diabetes (*n*, %)				0.015
No	7162 (91.7)	7042 (91.8)	120 (85.7)	
Yes	648 (8.3)	628 (8.2)	20 (14.3)	
Hypertension (*n*, %)				<0.001
No	5835 (74.4)	5756 (74.8)	79 (55.6)	
Yes	2006 (25.6)	1943 (25.2)	63 (44.4)	
Dyslipidemia (*n*, %)				0.581
No	3353 (42.8)	3296 (42.8)	57 (40.1)	
Yes	4488 (57.2)	4403 (57.2)	85 (59.9)	
Obesity (*n*, %)				0.376
No	4794 (64.1)	4716 (64.1)	78 (60.0)	
Yes	2688 (35.9)	2636 (35.9)	52 (40.0)	
Medication use (*n*, %)				<0.001
No	6919 (88.2)	6812 (88.5)	107 (75.4)	
Yes	922 (11.8)	887 (11.5)	35 (24.6)	
Nutraceutical intake (*n*, %)				0.044
No	6944 (88.7)	6810 (88.6)	134 (94.4)	
Yes	883 (11.3)	875 (11.4)	8 (5.6)	
MET (per day, mean ± SD)	109.82 ± 122.62	109.87 ± 122.68	107.17 ± 120.00	0.795
WC (cm, mean ± SD)	7661 ± 9.46	76.57 ± 9.46	78.67 ± 9.63	0.013
WHtR	5.52 ± 10.15	5.50 ± 10.15	6.83 ± 9.76	0.182
BMI (kg/m^2^, mean ± SD)	22.90 ± 3.36	22.89 ± 3.36	23.26 ± 3.25	0.203
FPG (mmol/L, mean ± SD)	5.25 ± 1.26	5.25 ± 1.25	5.40 ± 1.51	0.158
2h-PG (mmol/L, mean ± SD)	5.79 ± 2.25	5.79 ± 2.25	6.12 ± 2.52	0.088
SBP (mmHg, mean ± SD)	125.09 ± 20.87	124.90 ± 20.72	135.33 ± 25.97	<0.001
DBP (mmHg, mean ± SD)	78.24 ± 11.90	78.16 ± 11.85	82.56 ± 13.88	<0.001
TG (mmol/L, mean ± SD)	1.76 ± 1.57	1.75 ± 1.56	1.89 ± 1.92	0.324
CHOL (mmol/L, mean ± SD)	4.79 ± 1.32	4.79 ± 1.31	4.85 ± 1.55	0.64
HDL-C (mmol/L, mean ± SD)	1.45 ± 0.56	1.45 ± 0.56	1.41 ± 0.63	0.405
LDL-C (mmol/L, mean ± SD)	2.66 ± 1.18	2.66 ± 1.18	2.54 ± 1.30	0.239

Abbreviation: SD, standard deviation; MET, metabolic equivalent of task; WC, waist circumference; WHtR, waist-to-height ratio; BMI, body mass index; FPG, fasting plasma glucose; 2h-PG, 2-h postload glucose; SBP, systolic blood pressure; DBP, diastolic blood pressure, TG, triglyceride; CHOL, total cholesterol; HDL-C, high-density lipoprotein cholesterol; LDL-C, low-density lipoprotein cholesterol.

**Table 2 nutrients-14-00694-t002:** Distributions of scores for the DBI-16 components and the percentages of participants with each score.

Components	Score Range ^a^	Group		Distribution of Score (%)												*p* Value ^b^
			(−12)–(−11)	(−10)–(−9)	(−8)–(−7)	(−6)–(−5)	(−4)–(−3)	(−2)–(−1)	0	(1)–(2)	(3)–(4)	(5)–(6)	(7)–(8)	(9)–(10)	(11)–(12)	
Cereals	(−12)–(12)	Non-Ischemic stroke	0.6	0.8	1.4	2.2	4.8	1.7	16.0	1.1	18.2	8.8	5.6	2.9	36.0	0.296
		Ischemic stroke	0	1.4	0.7	0.7	5.6	1.4	16.2	2.8	13.4	5.6	7.7	2.1	42.3	
Vegetables	(−6)–(0)	Non-Ischemic stroke				4.2	26.3	31.5	38.0							0.249
		Ischemic stroke				1.4	23.2	35.9	39.4							
Fruits	(−6)–(0)	Non-Ischemic stroke				49.0	39.4	7.1	4.5							0.325
		Ischemic stroke				46.5	45.8	4.9	2.8							
Dairy	(−6)–(0)	Non-Ischemic stroke				91.3	6.2	2.3	0.3							0.167
		Ischemic stroke				88.7	5.6	4.9	0.7							
Soybeans	(−6)–(0)	Non-Ischemic stroke				31.0	12.8	10.6	45.6							0.065
		Ischemic stroke				24.6	14.1	16.9	44.4							
Red meats/products, Poultry/game	(−4)–(4)	Non-Ischemic stroke					5.2	28.3	19.8	15.6	31.1					0.004
		Ischemic stroke					12.0	32.4	16.2	14.1	25.4					
Fish/shrimps	(−4)–(0)	Non-Ischemic stroke					85.8	11.7	2.5							0.546
		Ischemic stroke					88.7	9.9	1.4							
Eggs	(−4)–(4)	Non-Ischemic stroke					49.4	34.1	13.3	1.4	1.8					0.896
		Ischemic stroke					49.3	36.6	12.0	0.7	1.4					
Cooking oils	(0)–(6)	Non-Ischemic stroke							36.2	22.4	12.4	29.0				<0.001
		Ischemic stroke							19.7	19.0	16.2	45.1				
Alcoholic beverages	(0)–(6)	Non-Ischemic stroke							93.8	3.2	1.3	1.7				0.009
		Ischemic stroke							91.5	1.4	4.2	2.8				
Addible sugar	(0)–(6)	Non-Ischemic stroke							98.9	0.7	0.1	0.3				0.915
		Ischemic stroke							99.3	0.7	0	0				
Salt	(0)–(6)	Non-Ischemic stroke							39.5	41.0	5.1	14.4				0.004
		Ischemic stroke							27.5	55.6	3.5	13.4				
Diet variety	(−12)–(0)	Non-Ischemic stroke	0	0.1	4.2	9.3	21.1	50.0	15.3							0.644
		Ischemic stroke	0	0	4.9	6.3	21.8	54.9	12.0							

^a^ Score range of total score is −60 to 44; ^b^
*p* value for chi-square test for the proportions of the scores for each food group.

**Table 3 nutrients-14-00694-t003:** Distribution of dietary quality and the percentages of participants with each category.

Diet Quality	Indicator	Score Range	Group	Mean ± SD	Distribution of Dietary Quality (%) ^a^				
					No Problem	Almost No Problem	Low Level Problem	Moderate Level Problem	High Level Problem
Under intake	LBS	0–60	Non-Ischemic stroke	22.68 ± 6.82	0	5.3	57.2	35.2	2.3
			Ischemic stroke	22.42 ± 7.09	0	5.6	62.0	29.6	2.8
Over intake	HBS	0–40	Non-Ischemic stroke	11.84 ± 6.68	2.7	36.0	43.8	16.6	0.9
			Ischemic stroke	13.30 ± 6.43	2.1	28.9	42.9	25.4	0.7
Overall unbalance	DQD	0–84	Non-Ischemic stroke	34.51 ± 8.35	0	1.4	50.7	44.0	3.9
			Ischemic stroke	35.72 ± 7.81	0	0	47.2	48.6	4.2

^a^ Distribution of the lower bound score (LBS): No problem: 0; Almost no problem: 1–12; Low level: 13–24; Moderate level: 25–36; High level: 37–60. Distribution of the higher bound score (HBS): No problem: 0; Almost no problem: 1–9; Low level: 10–18; Moderate level: 19–27; High level: 28–44. Distribution of the diet quality distance (DQD): No problem: 0; Almost no problem: 1–17; Low level: 18–34; Moderate level: 35–50; High level: 51–84.

**Table 4 nutrients-14-00694-t004:** Hazard ratios (HRs) and 95% confidence intervals (95%CIs) for ischemic stroke by diet quality indicators and DBI-16 components according to Cox regression models.

Indicators	No (*n*)	Cases (*n*)	Incident Density (Cases per 1000 PYs)	HR (95%CI) ^a^		
				Model 1	Model 2	Model 2
LBS ^b^						
Quartile 1 (Q1)	1761	34	2.78	1.00	1.00	1.00
Quartile 2 (Q2)	1907	41	3.06	1.04 (0.66–1.64)	1.13 (0.68–1.89)	1.13 (0.68–1.89)
Quartile 3 (Q3)	2008	29	2.07	0.78 (0.48–1.29)	0.79 (0.44–1.41)	0.76 (0.43–1.36)
Quartile 4 (Q4)	2165	38	2.46	0.92 (0.57–1.46)	0.86 (0.46–1.59)	0.84 (0.45–1.56)
HBS ^c^						
Quartile 1 (Q1)	1619	14	1.19	1.00	1.00	1.00
Quartile 2 (Q2)	2042	37	2.57	2.24 (1.21–4.15) *	2.38 (1.12–5.05)*	2.38 (1.12–5.06) *
Quartile 3 (Q3)	2168	44	2.89	2.48 (1.36–4.53) **	2.38 (1.14–5.00)*	2.39 (1.14–5.01) *
Quartile 4 (Q4)	2012	47	3.43	3.12 (1.72–5.68) ***	3.15 (1.50–6.63)**	3.31 (1.57–6.97) **
DQD ^d^						
Quartile 1 (Q1)	1853	26	1.98	1.00	1.00	1.00
Quartile 2 (Q2)	1797	32	2.50	1.28 (0.76–2.15)	1.34 (0.75–2.37)	1.33 (0.75–2.36)
Quartile 3 (Q3)	2166	36	2.36	1.29 (0.78–2.13)	1.11 (0.62–2.01)	1.13 (0.63–2.04)
Quartile 4 (Q4)	2025	48	3.44	1.99 (1.23–3.23) **	2.19 (1.24–3.86) **	2.26 (1.28–4.00) **
Cereals						
Score 0	1252	23	2.61	1.00	1.00	1.00
Score (−12)–(−7)	220	3	1.90	0.76 (0.23–2.51)	1.06 (0.31–3.62)	1.03 (0.30–3.53)
Score (−6)–(−1)	680	11	2.26	0.85 (0.41–1.74)	0.69 (0.29–1.65)	0.65 (0.27–1.55)
Score (1)–(6)	2192	31	2.01	0.74 (0.43–1.26)	0.55 (0.29–1.04)	0.54 (0.28–1.02)
Score (7)–(12)	3497	74	3.02	1.08 (0.67–1.72)	0.95 (0.56–1.61)	0.94 (0.55–1.59)
Vegetables						
Score 0	2982	56	2.67	1.00	1.00	1.00
Score (−6)–(−1)	4859	86	2.52	0.98 (0.70–1.37)	1.02 (0.68–1.53)	1.05 (0.70–1.57)
Fruits						
Score 0	354	4	1.62	1.00	1.00	1.00
Score (−6)–(−1)	7487	138	2.62	1.75 (0.65–4.73)	1.88 (0.59–6.04)	1.95 (0.61–6.28)
Dairy						
Score 0	25	1	5.84	1.00	1.00	1.00
Score (−6)–(−1)	7816	141	2.57	0.54 (0.07–3.82)	0.31 (0.04–2.30)	0.31 (0.04–2.26)
Soybeans						
Score 0	3575	63	2.52	1.00	1.00	1.00
Score (−6)–(−1)	4266	79	2.62	1.09 (0.78–1.53)	1.04 (0.69–1.57)	1.03 (0.68–1.54)
Meats						
Score 0	1544	23	2.12	1.00	1.00	1.00
Score (−4)–(−1)	2645	63	3.4	1.66 (1.03–2.68) *	1.13 (0.67–1.90)	1.08 (0.64–1.82)
Score (1)–(4)	3652	56	2.17	0.95 (0.59–1.55)	0.64 (0.37–1.12)	0.63 (0.36–1.09)
Fish/shrimps						
Score 0	194	2	1.47	1.00	1.00	1.00
Score (−4)–(−1)	7647	140	2.60	1.76 (0.44–7.10)	2.02 (0.28–14.50)	1.98 (0.28–14.20)
Eggs						
Score 0	1042	17	2.37	1.00	1.00	1.00
Score (−4)–(−1)	6547	122	2.64	1.01 (0.61–1.68)	0.78 (0.45–1.35)	0.79 (0.46–1.37)
Score (1)–(4)	252	3	1.69	0.65 (0.19–2.21)	0.52 (0.12–2.25)	0.50 (0.12–2.20)
Cooking oils						
Score 0	2814	28	1.39	1.00	1.00	1.00
Score (1)–(6)	5027	114	3.26	2.60 (1.72–3.94) ***	2.96 (1.75–5) ***	3.00 (1.77–5.07) ***
Alcoholic beverages						
Score 0	7353	130	2.51	1.00	1.00	1.00
Score (1)–(6)	488	12	3.64	1.62 (0.90–2.93)	1.30 (0.60–2.80)	1.35 (0.62–2.93)
Addible sugar						
Score 0	7757	141	2.59	1.00	1.00	1.00
Score (1)–(6)	84	1	1.74	0.81 (0.11–5.80)	0.84 (0.12–6.00)	0.81 (0.11–5.81)
Salt						
Score 0	3079	39	1.76	1.00	1.00	1.00
Score (1)–(6)	4762	103	3.13	2.04 (1.41–2.96) ***	1.98 (1.29–3.02) **	2.03 (1.33–3.10) **
Dietary variety						
Score 0	1195	17	1.97	1.00	1.00	1.00
Score (−12)–(−7)	337	7	2.85	1.58 (0.65–3.85)	5.24 (1.66–16.50) **	5.40 (1.70–17.20) **
Score (−6)–(−1)	6309	118	2.68	1.65 (0.99–2.76)	1.66 (0.94–2.95)	1.69 (0.95–3.01)

^a^ Model 1: Adjusted for age only; Model 2: Model 1 + additionally adjusted for sex, area, ethnic group, education level, marriage status, economic level, smoking, diabetes, hypertension, dyslipidemia, and obesity status; Model 3: Model 2 + additionally adjusted for medication use and nutraceutical intake. ^b^ Quartile levels for the lower bound score (LBS): Q1, score 0–18; Q2, score 19–22; Q3, score 23–27; Q4, score 28–52. ^c^ Quartile levels for the higher bound score (HBS): Q1, score 0–6; Q2, score 7–12; Q3, score 13–17; Q4, score 18–33. ^d^ Quartile levels for the diet quality distance (DQD): Q1, score 9–29; Q2, score 30–34; Q3, score 35–40; Q4, score 41–68. * *p* < 0.05; ** *p* < 0.01; *** *p* < 0.001.

## Data Availability

The datasets for this manuscript will be made available upon request pending, further inquiries can be directed to the corresponding author Liu Tao, liutaombs@163.com and Na Wang, na.wang@fudan.edu.cn.

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
