# Peer review of "Unfavorable Dietary Quality Contributes to Elevated Risk of Ischemic Stroke among Residents in Southwest China: Based on the Chinese Diet Balance Index 2016 (DBI-16)"

_nutrients, 2022, doi:10.3390/nu14030694_

Round 1
Reviewer 1 Report
This work describe for the first time the relationship between ischemic stroke development and dietary change in Chinese population. I think that this work is well written and appropriate described
Author Response
Thank you very much for your review and comments.
Reviewer 2 Report
To:
Editorial Board
Nutrients
Title: “Unfavorable dietary quality contributes to elevated risk of ischemic stroke among residents in Southwest China: based on the Chinese Diet Balance Index 2016 (DBI-16)”
Dear Editor,
I read this manuscript and I think that:
- The use of a questionnaire is a limitation of the study. Please discuss such a point in a dedicated limitation section. Please provide.
- Inclusion and exclusion criteria should be discussed and better described. Please provide.
- The authors should also discuss the role of nutraceuticals in clinical practice. They can consider and discuss the paper from Scicchitano P et al. Journal of Functional Foods 2014;6:11-32.
- Pharmacological treatments should be described and included in the final analysis.
- Comorbities should be described and included in the final regression analysis.
Author Response
Dear reviewer,
Thank you for taking the time and effort to review our manuscript (ID: 825501) entitled “Unfavorable dietary quality contributes to elevated risk of ischemic stroke among residents in Southwest China: based on the Chinese Diet Balance Index 2016 (DBI-16) ”. We appreciate your constructive comments and have revised the manuscript carefully in order to meet your requirements. We have uploaded the revised marked manuscript with all the changes highlighted in Red. Appended to this letter is our point-by-point response to the comments raised by the editors and reviewers. The comments are listed and our responses are given directly afterward.
1.The use of a questionnaire is a limitation of the study. Please discuss such a point in a dedicated limitation section. Please provide.
Response:
Thank you for you suggestion. The limitations of the use of questionnaire has been discussed in the limitation section (page 16-17, line 554-559):
“Firstly, dietary habits and socioeconomic characteristics were collected based on individual self-report, which might lead to the recall bias. Food frequency questionnaire (FFQ) has considered as a convenient and widely-used dietary assessment tool, but this method is subject to the less accuracy of quantification of food portions than the method of weighing, which might make some measurement errors.”
2.Inclusion and exclusion criteria should be discussed and better described. Please provide.
Response:
Detailed descriptions of inclusion and exclusion criteria have been added in the section “2. Methods-2.1 Study design and participants” (page 2, line 74-91):
“The Guizhou Population Health Cohort Study (GPHCS) is one of few large population-based prospective cohort studies in Southwest China, which was established from 12 areas (5 urban districts and 7 rural counties) in Guizhou Province between November 20, 2010 and December 19, 2012 . A multistage proportional stratified cluster sampling method was used to obtain a representative sample of general population in Guizhou Province. The inclusive criteria were as follows: (1) age of 18 years or above; (2) living in the study regions for more than 6 months and having no plan to move out; (3) completing survey questionnaire and blood sampling; (4) signing the written informed consent. A total of 9280 local residents were enrolled into the cohort. All participants were followed up for major chronic diseases and vital status through a repeated investigation conducted during 2016 to 2020. All deaths were confirmed by the record from Death Registration Information System and Basic Public Health Service System.Ethic approval was obtained from the ethic review board of Guizhou Province (No.S2017-02). All participants provided the written informed consent at enrollment.
In this study, we excluded participants with history of ischemic stroke, haemorrhagic stroke, myocardial infarction or other cardiovascular diseases, missing data of diet consumption at baseline, loss to follow-up and death, leaving 7841 participants for the analyses (Figure 1).”
3.The authors should also discuss the role of nutraceuticals in clinical practice. They can consider and discuss the paper from Scicchitano P et al. Journal of Functional Foods 2014;6:11-32.
Pharmacological treatments should be described and included in the final analysis.
Comorbidities should be described and included in the final regression analysis.
Response:
Thank you for your suggestions and we added this analyses in the revised manuscript.
(1) The definitions of several comorbidities (including diabetes, hypertension, dyslipidemia and obesity), medication use, and nutraceuticalintake have been added in the section “Methods- 2.5 Other variable collection”:
â‘ The definitions of diabetes, hypertension, dyslipidemia and obesity: page 4, line 176-181
â‘¡ The definitions of medication use: page 4, line 151-152
â‘¢ The definitions of nutraceutical intake: page 4, line 153-155
Considering the relative backward in economic development in Guizhou Province and the less demand in nutraceuticals among local residents, we just investigated the intake of several common nutraceuticals and foods with health-care functions, including vitamin, minerals, wine and tea, which may play a protective role on the cardiovascular system.
(2) The distributions of these comorbidities, the status of medication use and nutraceuticalintake at baseline has been added in Table 1(page 5-7).
(3) In the cox regression analyses, we adjusted for comorbidities in Model 2, and additionally adjusted for medication use + nutraceuticalintake in Model 3. and the corresponding results have been revised in Table 3(page 10-11).
(4) Subgroup analyses have been conducted among those with different status of several comorbidities, medication use, and nutraceuticalintake, and the corresponding results have been performed in Figure 3 (page 12-13).
(5) Corresponding discussions were performed in the discussion section (page 16, line 535-545):
“Given that the potential effects of comorbidities, some medications for metabolic diseases and some nutraceuticals, which may promote or decelerate the progress of cardiovascular diseases,36 37 we performed the same analyses in those with different status of these conditions. The risk effects of excessive food intake (evaluated by HBS) and unbalance food intake were more evident in those with hypertension history and taking medications for metabolic diseases, indicating that people with a high risk of stroke should pay more attention to the balance of food types and daily intake, reasonable diet is also one of the major measures to prevent hypertension.16 However, we failed to observe any associations of ischemic stroke with diet quality among those with diabetes or intaking nutraceuticals, perhaps due to smaller sample size in those subgroups.”
